# Emergence of Drift Variants That May Affect COVID-19 Vaccine Development and Antibody Treatment

**DOI:** 10.3390/pathogens9050324

**Published:** 2020-04-26

**Authors:** Takahiko Koyama, Dilhan Weeraratne, Jane L. Snowdon, Laxmi Parida

**Affiliations:** 1TJ Watson Research Center, IBM, Yorktown Heights, NY 10598, USA; 2Center for Artificial Intelligence, Research, and Evaluation, IBM, Cambridge, MA 02142, USA

**Keywords:** SARS-CoV-2, COVID-19, genomic drift, variant, immune escape, vaccine, antibody, spike protein, convalescent plasma

## Abstract

New coronavirus (SARS-CoV-2) treatments and vaccines are under development to combat COVID-19. Several approaches are being used by scientists for investigation, including (1) various small molecule approaches targeting RNA polymerase, 3C-like protease, and RNA endonuclease; and (2) exploration of antibodies obtained from convalescent plasma from patients who have recovered from COVID-19. The coronavirus genome is highly prone to mutations that lead to genetic drift and escape from immune recognition; thus, it is imperative that sub-strains with different mutations are also accounted for during vaccine development. As the disease has grown to become a pandemic, B-cell and T-cell epitopes predicted from SARS coronavirus have been reported. Using the epitope information along with variants of the virus, we have found several variants which might cause drifts. Among such variants, 23403A>G variant (p.D614G) in spike protein B-cell epitope is observed frequently in European countries, such as the Netherlands, Switzerland, and France, but seldom observed in China.

## 1. Introduction

In late 2019, a new coronavirus, SARS-CoV-2, causing acute respiratory distress syndrome, was first reported in Wuhan, China. Despite a lockdown of the city, the number of patients increased exponentially, while in parallel the virus spread across the globe. The World Health Organization (WHO) declared a pandemic on 11 March 2020. Currently, no treatments or vaccines are scientifically proven to be effective against the virus. Safe and effective vaccines for SARS-CoV-2 are urgently needed to mitigate the pandemic. To that end, a clinical trial of mRNA-1273 with full spike protein as an antigen started on 8 March 2020 [1].

Pharmaceutical companies are currently investigating repurposed compounds from other infections as potential treatments for COVID-19. For instance, lopinavir and ritonavir are both HIV protease inhibitors; however, the derived treatment benefit was dubious in a lopinavir–ritonavir clinical trial that was recently reported [2]. Remdesivir, an RNA polymerase inhibitor originally intended to treat Ebola virus, appears to have in vitro activity against SARS-CoV-2 [3] and preliminary clinical activity [4]. Additionally, convalescent immunoglobulins derived from recovering patients are currently being investigated as a potential treatment for the disease [5]. Until a widely available, efficient vaccine exists, these treatments are the best hope to reduce mortality.

Typically, surface proteins outside of the viral virion are selected for antigens so that antibodies generated from a vaccine-trained B-cell can bind to the virus for neutralization. In addition to the B-cell epitope requirement, the antigens must generate antigenic peptides, which bind to the major histocompatibility complex (MHC) molecules to be presented. By presenting a peptide, a B-cell can become stimulated by a helper T-cell and become a plasma cell to generate antibodies. A fraction of stimulated B-cells are transferred to the germinal center, where they are further enhanced from random somatic mutagenesis induced by activation-induced deaminase (AID) allowing stronger binding to the antigen. Therefore, the resulting antibodies have differences in binding epitope and protein sequences in variable antibody regions. The antigens introduced as vaccines need to account for current major sub-strains to prevent potential escape from immune recognition.

Genetic drift takes place when the occurrence of alleles or variant forms of a gene increase or decrease over time [6]. Genetic drift is measured by the changes in allele frequencies and continues until one of two possible events occurs: the involved allele is lost by a population or the involved allele is the only allele present in a population at a particular locus. Genetic drift may cause a new population to be genetically distinct from the original population. This study’s objective is to interrogate currently identified sub-strains of SARS-CoV-2 and identify genetic drifts and potential immune recognition escape sites that would be integral for the development of a successful vaccine.

## 2. Materials and Methods

Predicted B-cell and T-cell epitopes were obtained from results of assays performed for SARS-CoV and sequence alignments between SARS-CoV and SARS-CoV-2 from the recent work by Grifoni et al. [7]. The sequence identity and similarity of spike protein between the strains was 76.3% and 87.0%, respectively, after running Needle pairwise alignment [8]. As shown in Figure 1, the spike protein sequences of SARS-CoV and SARS-CoV-2 have high similarity in the regions of interest, which are colored in blue. For instance, in the segment ranging 601–640, 32 out of 41 (78%) residues are identical, 5 out of 41 (12%) residues are similar, and 4 out of 41 (10%) residues are dissimilar. Therefore, we assume that epitopes predicted from SARS-CoV results are reliable.

In total, 615 variant data files in general feature format (GFF3) were downloaded from China’s National Genomics Data Center (NGDC) (https://bigd.big.ac.cn/ncov/release_genome?lang=en) on 20 March 2020. They provide the variant information from the Global Initiative on Sharing All Influenza Data (GISAID) [9], GenBank, NGDC Genome Warehouse, and National Microbiology Data Center (NMDC). Sample information is provided in Appendix A. Samples with hyper mutations and large gaps were considered to be of low quality and were discarded from the analysis. The GFF3 files were processed to extract sample information, including genome accession number, geographic location, sample collection date, coordinate information, base changes, genes, amino acid changes, and variant types, and were then organized into a database. We searched for variants located within each predicted epitope and then tabulated these, as shown in Table 1. Additionally, country-based statistics of the prevalence of 23403A>G variant (p.D614G) were generated, as shown in Table 2.

## 3. Results

Twelve distinct variants were found within B-cell epitopes of spike protein (S), nucleocapsid protein (N), and membrane protein (M), respectively, as listed in Table 1. Also, twenty-one distinct variants were identified in T-cell epitopes. 

Among the twelve variants in the B-cell epitopes, 23403A>G variant (p.D614G) in one of the epitopes in spike protein between residue 601 and 640 stands out, with 175 samples in 615 total samples. The variant is located in the middle of that epitope and the amino acid change in the 23403A>G variant (p.D614G) involves a change of large acidic residue D (aspartic acid) into small hydrophobic residue G (glycine). Such large differences in both size and hydrophobicity in the middle of the epitope would compromise the binding affinity to antibodies trained by vaccines with wild-type spike protein. Most of the samples with the variant were collected in Europe, in particular the Netherlands (66 out of 112), Switzerland (29 out of 30), and France (21 out of 32), as shown in Table 2. In these countries, the majority of infected patients possess the variant; therefore, vaccine design and convalescent plasma antibody treatment might require further considerations to accommodate the drift.

## 4. Discussion

The immunogenicity of SARS-CoV-2 proteins can be extrapolated from very close sequence homology to SARS-CoV-1. Five regions of immunodominance, including residues from 601 to 640 in the spike protein, have been reported from SARS-CoV-1 and 78% homology is observed in that region with SARS-CoV-2. Notably, the D614 residue is conserved between the two SARS strains. A spike glycoprotein peptide encompassing residues 604–625 derived from a convalescent SARS-CoV-1 patient was successfully able to elicit humoral immune response and prevent infection in non-human primates, underscoring the immunogenic importance of this region [10]. 

In addition to the Netherlands, Switzerland, and France, our data indicate that the D614G sub-strain is frequently detected in Brazil, Finland, and Belgium. However, given the small sample size, it is hard to ascertain whether D614G is the dominant strain in these countries. A recent report corroborated our findings of high prevalence of D614G in Europe [11]. Within the analyzed patient cohort, the variant was first observed in EPI_ISL_406862, collected on January 28, 2020, in a sample from Germany. Subsequently, the variant was detected in EPI_ISL_412982, collected on February 7, 2020, in a sample from Wuhan, China. Notably, these two samples do not share common variants besides p.D614G. It is unclear whether the variant emerged in China and disseminated to Europe or this variant emerged independently in China and Europe. Intriguingly, in our data the D614G variant was detected only in 2 out of 151 Chinese patients analyzed.

The reports of reinfection and relapse of COVID-19 disease suggest that eliciting an effective and lasting host immune response to facilitate viral clearance can be a challenge, at least in some patients. As viruses mutate during replication, host antibodies generated in the earlier phase of the infection may not be as effective later on [12]. While the precise effects of glycine in lieu of aspartic acid in residue 614 on immunogenicity and virus neutralization potential are currently unknown, it may confer conformational changes, which may affect binding. Although a single amino acid change may not affect binding to antibodies, a new variant in the same epitope may emerge as the viral genome evolves. In fact, we observed p.V615L in our data set, suggesting that a second or a third hit could occur in the same epitope while a vaccine is being developed. Therefore, it is imperative that currently known variants of COVID-19, as well as new variants that may occur as the viral genome mutates, are carefully considered in the design of a vaccine.

## 5. Conclusions

The highly prevalent 23403A>G (p.D614G) variant in the European population may cause antigenic drift, resulting in vaccine mismatches that offer little protection to that group of patients. Innovative vaccine design methods, including using highly conserved internal epitopes, recombinant proteins spanning epitopes, or pooling multiple vaccines, will be required to combat the inherent antigenic drift. Consideration of drift variants in SARS-CoV-2 will offer cross-protection across different sub-strains and obviate the need for reformulation of the vaccine for each distinct sub-strain. Additionally, consideration of drift variants in convalescent immunoglobulin treatment strategies will also result in better patient outcome. In conclusion, consideration of antigenic drift in the different sub-strains of the virus is imperative in the design of a “one size fits all” universal vaccine to offer protection against the deadliest outbreak in this century.

## Figures and Tables

**Figure 1 pathogens-09-00324-f001:**
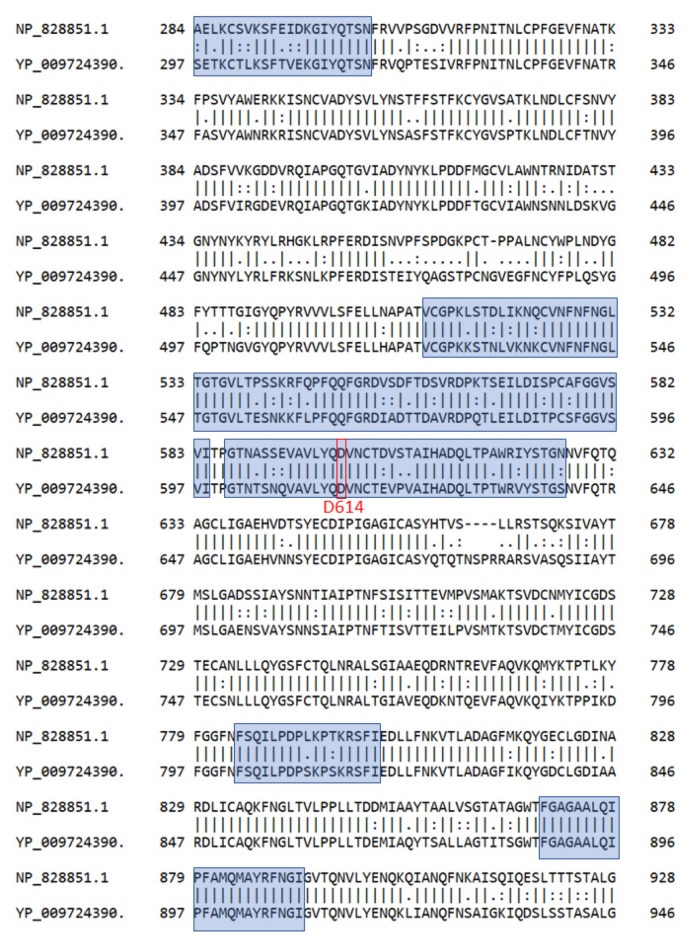
Pairwise sequence alignments of spike protein (S) between SARS-CoV (NP_828851.1) and SARS-CoV-2 (YP_009724390). Similarities in the predicted B-cell epitopes in blue are high. D614 residue is marked with a red rectangle.

**Table 1 pathogens-09-00324-t001:** SARS-CoV2 variants that occur in the predicted epitopes in spike protein (S), nucleocapsid protein (N), and membrane protein (M).

Cell Type	Epitope	Protein	Residues	Amino Acid Change	Base Change	Number of Samples
**B-CELL**	GTNTSNQVAVLYQDVNCTEVPVAIHADQLTPTWRVYSTGS	S	601–640	p.V615L	23405G>C	1
**B-CELL**	GTNTSNQVAVLYQDVNCTEVPVAIHADQLTPTWRVYSTGS	S	601–640	p.D614G	23403A>G	175
**B-CELL**	FSQILPDPSKPSKRSFIE	S	802–819	p.F817L	24011T>C	1
**B-CELL**	FSQILPDPSKPSKRSFIE	S	802–819	p.P812S	23996C>T	1
**B-CELL**	FGAGAALQIPFAMQMAYRFNGI	S	888–909	p.M902fs	24268del	1
**B-CELL**	MADSNGTITVEELKKLLEQWNLVI	M	1–24	p.D3G	26530A>G	5
**B-CELL**	RPQGLPNNTASWFTALTQHGK	N	41–61	p.P46S	28409C>T	1
**B-CELL**	NNNAATVLQLPQGTTLPKGF	N	153–172	p.A156S	28739G>T	2
**B-CELL**	NKHIDAYKTFPPTEPKKDKKKKTDEAQPLPQRQKKQPTVTLLPAADM	N	355–401	p.E378Q	29405G>C	1
**B-CELL**	NKHIDAYKTFPPTEPKKDKKKKTDEAQPLPQRQKKQPTVTLLPAADM	N	355–401	p.K373N	29392G>T	1
**B-CELL**	NKHIDAYKTFPPTEPKKDKKKKTDEAQPLPQRQKKQPTVTLLPAADM	N	355–401	p.K370N	29383G>T	1
**B-CELL**	NKHIDAYKTFPPTEPKKDKKKKTDEAQPLPQRQKKQPTVTLLPAADM	N	355–401	p.P365S	29366C>T	1
**T-CELL**	QPFLMDLEGKQGN	S	173–185	p.G181V	22104G>T	1
**T-CELL**	TRFQTLLALHRSYLTPGDSSSGW	S	236–258	p.S254F	22323C>T	2
**T-CELL**	TRFQTLLALHRSYLTPGDSSSGW	S	236–258	p.S247R	22303T>A/G	3
**T-CELL**	TRFQTLLALHRSYLTPGDSSSGW	S	236–258	p.L241_A243del	22281_22289del	1
**T-CELL**	TRFQTLLALHRSYLTPGDSSSGW	S	236–258	p.Q239K	22277C>A	6
**T-CELL**	NLDSKVGGNYNYLYRLFR	S	440–457	p.F456fs	22928del	1
**T-CELL**	YLYRLFRKSNLKPFERDI	S	451–468	p.K458R	22935A>G	1
**T-CELL**	YLYRLFRKSNLKPFERDI	S	451–468	p.F456fs	22928del	1
**T-CELL**	TECSNLLLQYGSFCTQL	S	747–763	p.L752F	23816C>T	1
**T-CELL**	VKQIYKTPPIKDFGGFNF	S	785–802	p.F797C	23952T>G	1
**T-CELL**	VKQIYKTPPIKDFGGFNF	S	785–802	p.T791I	23934C>T	1
**T-CELL**	DSLSSTASALGKLQDVV	S	936–952	p.S943T	24390G>C	4
**T-CELL**	DSLSSTASALGKLQDVV	S	936–952	p.S943R	24389A>C	3
**T-CELL**	DSLSSTASALGKLQDVV	S	936–952	p.T941A	24383A>G	1
**T-CELL**	DSLSSTASALGKLQDVV	S	936–952	p.S940F	24381C>T	2
**T-CELL**	DSLSSTASALGKLQDVV	S	936–952	p.S939F	24378C>T	2
**T-CELL**	RLNEVAKNL	S	1185–1193	p.A1190G	25131C>G	1
**T-CELL**	RLNEVAKNL	S	1185–1193	p.N1187K	25123T>A	1
**T-CELL**	RIFTIGTVTLKQGEI	ORF3a	6–20	p.F8L	25414T>C	1
**T-CELL**	GMSRIGMEV	N	316–324	p.M322I	29239G>T	1
**T-CELL**	MEVTPSGTWL	N	322–331	p.S327L	29253C>T	1

**Table 2 pathogens-09-00324-t002:** Statistics of 23403A>G variant (p.D614G) in spike protein observed by country.

Country	Variant Count	Total Count
Netherlands	66	112
Switzerland	29	30
France	21	32
United Kingdom	12	30
USA	9	123
Brazil	8	13
Belgium	7	8
Finland	6	7
Portugal	2	2
Italy	2	6
Ireland	2	3
Germany	2	9
Denmark	2	2
China	2	151
Russia	1	1
Mexico	1	1
Luxemburg	1	1
Georgia	1	3
Chile	1	7

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
