# Peer review of "Emergence of Drift Variants That May Affect COVID-19 Vaccine Development and Antibody Treatment"

_pathogens, 2020, doi:10.3390/pathogens9050324_

Round 1

Reviewer 1 Report

In their manuscript Koyama et al aimed is to identify drift variants and potential immune escape sites. Through their analysis of viral sequences, the authors identified the variant D614G in spike as being highly prevalent particularly in samples from the Netherlands, Switzerland and France. The identified residue lies on a recessed surface of spike domain D. The authors make the claim that this variant will require considerations to the use of current vaccine strategies and the use of convalescent plasma to accommodate this drift. However, a single amino acid change on a predicted epitope not currently known for immunodominance, is highlyunlikely to affect the polyclonal antibody reactivity of current vaccination strategies or treatment with convalescent plasma. Though I believe that the work is valid, the lack of context of this work within what is known about coronavirus spike antibodies makes many of the conclusions drawn from this work overstated. As minor comments, the manuscript would benefit from language and grammar editing and the introduction is repetitive.

Author Response

Introduction: As suggested, repetitive statement are removed, and content is streamlined (line 34 and lines 56-65). We have also introduced relevant new references (line 39).

Conclusion: Thank you for the suggestion to be cautious in the interpretation of results. We have referenced that the 604 – 625 region in the spike protein is highly immunogenic based on sequence homology to other coronaviruses and also referenced that convalescent plasma from this region has been efficacious in a non-primate model (lines 158-164). We have duly noted that a single amino acid by itself may not affect antibody recognition, but in conjunction with other variants in the same epitope warrants consideration during vaccine development (lines 190-195)

Reviewer 2 Report

This manuscript explores the occurrence of several drift variants of SARS-CoV-2. One of the variants 23403A>G variant (p.D614G) which is prevalent in France, Netherlands, and Switzerland has amino acid change in the B-cell epitope of the spike protein, which might affect the affinity to antibodies to the wild type. It would be interesting to further investigate the biological significance of these genetic drift. 

The study is well designed and executed. The manuscript is well written but the Materials and Methods section is not described in detail.  

The findings of this study is significant and add one more piece to the puzzle. It provides crucial data to the teams developing and testing the urgently needed vaccine to combat the COVID-19 pandemic. And, the genetic drift data could be beneficial during the convalescent immunoglobulin treatment. 

Author Response

Materials and Methods: As suggested we have included new content in this section to describe the methods in more detail (lines 74-126).

We appreciate the opportunity to revise our manuscript. We believe that our manuscript is of interest to the readership of Pathogens given the expediency in developing drugs and vaccines to fight the COVID-19 pandemic. Thank you for the consideration.